# Enhancing Video Representation Learning with Temporal Differentiation

Siyi Chen[1], Minkyu Choi[1], Zesen Zhao[1], Kuan Han[1], Qing Qu[1], and Zhongming Liu[2]

[1]Department of Electrical Engineering & Computer Science, University of Michigan
[2]Department of Biomedical Engineering, University of Michigan

Taking inspiration from physical motion, we present a new self-supervised dynamics learning strategy for videos: **Vi**deo Time-**Di**fferentiation for Instance **Di**scrimination (ViDiDi). ViDiDi is a simple and data-efficient strategy, readily applicable to existing self-supervised video representation learning frameworks based on instance discrimination. At its core, ViDiDi observes different aspects of a video through various orders of temporal derivatives of its frame sequence. These derivatives, along with the original frames, support the Taylor series expansion of the underlying continuous dynamics at discrete times, where higher-order derivatives emphasize higher-order motion features. ViDiDi learns a single neural network that encodes a video and its temporal derivatives into consistent embeddings following a balanced alternating learning algorithm. By learning consistent representations for original frames and derivatives, the encoder is steered to emphasize motion features over static backgrounds and uncover the hidden dynamics in original frames. Hence, video representations are better separated by dynamic features. We integrate ViDiDi into existing instance discrimination frameworks (VICReg, BYOL, and SimCLR) for pretraining on UCF101 or Kinetics and test on standard benchmarks including video retrieval, action recognition, and action detection. The performances are enhanced by a significant margin without the need for large models or extensive datasets.

## 1. Introduction

Learning video representations is central to various aspects of video understanding, such as action recognition [1, 2], video retrieval [3, 4], and action detection [5, 6]. While supervised learning requires expensive video labeling [7], recent works highlight the strengths of self-supervised learning (SSL) from unlabeled videos [8–10] with a large number of training videos.

One popular strategy for SSL on video representations uses instance discrimination objectives, such as SimCLR [11], initially demonstrated for images [11–15] and then generalized to videos [9, 16, 17]. In images, models learn to pair together latent representations for the same instance under different augmented views. Such models effectively filter out lower-level details and generate abstract representations useful for higher-level tasks. When adapting this approach to videos, previous methods often use clips from different times as separate views of the same video, treating time as an additional spatial dimension. However, learning consistent representations across clips may cause models to prioritize static content (e.g., background scenes) over dynamic features (e.g., motion, action, and interaction), which are often essential to video understanding [1, 2, 7, 18–21]. Despite some pretext tasks handling time in distinction from space [3, 8, 9, 16, 17, 22, 23], they are not generalizable or principled. In contrast, we utilize the unique role of time in "unfolding" continuous real-world dynamics. We provide a more detailed discussion on prior arts in Appendix A.

In this paper, we introduce a generalizable and data-efficient method for improved dynamics learning, applicable to self-supervised video representation learning through instance discrimination. Central to our approach, we view a video not just as a sequence of discrete frames but as a continuous and dynamic process. We use the Taylor series expansion to express this continuous pro-

Second Conference on Parsimony and Learning (CPAL 2025).

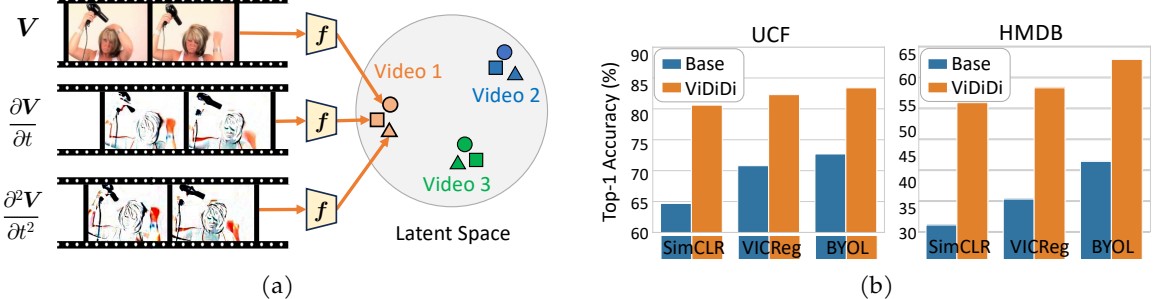

Figure 1: **Method Overview**. (a) The ViDiDi method evaluates temporal derivatives of video clips through Taylor expansion, uses the same encoder to embed them into the latent space, and converges their representations for the same video while diverging those from different videos. (b) Pretraining via ViDiDi enhances existing instance discrimination methods significantly on action recognition.

cess as a weighted sum of its temporal derivatives at each frame. Using the physical motion as a metaphor, the zeroth-order derivative represents each frame itself, analogous to position. The first-order derivative captures the immediate motion between frames, analogous to velocity. The second-order derivative reveals the rate of change in this motion, analogous to acceleration. We train models to align representations for the original video and its first and second-order temporal derivatives, such that the learned representations encode the underlying dynamics more consistently among different orders of temporal differentiation (fig. 1a). This method mirrors our intuitive perception of motion in the physical world, where a holistic understanding of position, velocity, and acceleration altogether helps us relate an object's trajectory to the underlying laws of physics.

Herein, we refer to this approach as Video Time-Differentiation for Instance Discrimination (ViDiDi). We have implemented ViDiDi across three SSL methods using instance discrimination, including VICReg [14], BYOL [12], and SimCLR [11]. ViDiDi is not merely an image processing approach. It introduces new perspectives on understanding dynamic data including but not limited to videos, and uses a balanced alternating learning strategy to guide the learning process. We tested ViDiDi with different encoder architectures as well as different learning objectives. Pretrained on UCF101 or Kinetics, our method demonstrates excellent generalizability and data efficacy on standard benchmarks including action recognition (fig. 1b) and video retrieval.

Our contributions are summarized as follows:

- Introducing *a new view* for representing *continuous dynamics* with different orders of temporal derivatives using *Taylor series expansion* inspired by physics;

- Proposing an general *self-supervised dynamics learning framework* that learns representations *consistent among different temporal derivatives* with a *balanced alternating learning strategy*, applicable to multiple existing self-supervised learning approaches;

- Demonstrating the *data efficiency* and substantial *performance gains* on common video representation learning tasks of the proposed method, and analyzing the learned dynamic features via *attention and subspace clustering visualization*.

## 2. Approach

In this section, we first describe the mathematical framework and explain our intuition through a thought experiment grounded in physics (section 2.1). Then describe in detail how we incorporate temporal differentiation with a balanced alternating learning strategy (section 2.2) into two-stream self-supervised learning methods. We use SimCLR [11] as a specific example for constructing and training a ViDiDi-based model while referring readers to other implementations in appendix C.

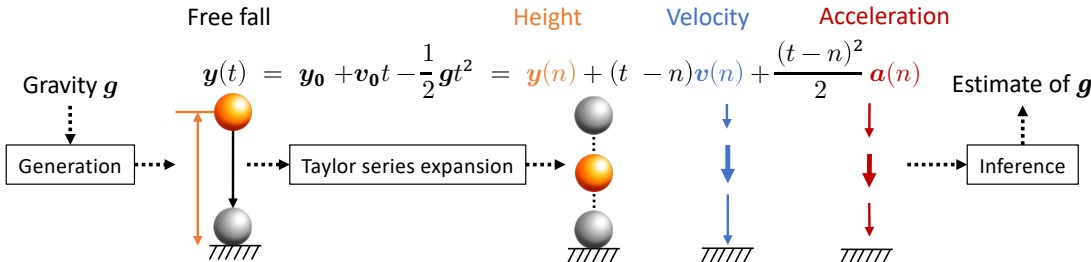

Figure 2: **A thought experiment on physical motion.** The Taylor series expansion projects the dynamic process of a free fall motion onto three views expressed in terms of the height, velocity, and acceleration. The reverse inference of the common causes of the height, velocity, and acceleration leads to the encoding of the gravity $g$ - the only variable pertaining to all three views, instead of unrelated static latents, $y_0$ and $v_0$.

### 2.1. Recover Hidden Views of Video Sequence

**Taylor series expansion of videos.** We view one video not just as a sequence of $N$ discrete frames $[y(n)]$, $n = 1, 2, \ldots, N$, but as a continuous dynamic process $y(t)$. We assume that this process is caused by latent factors, $s$ and $z$, which account for the static and dynamic aspects of the video through an unknown generative model:

$$y(t) = g_1(s) + g_2(z, t) \tag{1}$$

where $g_1(s)$ generates a static environment and $g_2(z, t)$ generates a dynamic process situated in the environment.

Although $y(t)$ is not observed, we can approximate $y(t)$ for any time $t$ around specific time $n$ through the Taylor series expansion:

$$y(t) = y(n) + (t - n)\frac{\partial y}{\partial t}(n) + (t - n)^2 \frac{1}{2!}\frac{\partial^2 y}{\partial t^2}(n) + \ldots \tag{2}$$

Here, the original frame $y(n)$ can be viewed as the zeroth order derivatives. Its contributions to $y(t)$ do not incorporate a continuous time variable $t$. $\frac{\partial y}{\partial t}(n) = \frac{\partial g_2}{\partial t}(z, n)$ and $\frac{\partial^2 y}{\partial t^2}(n) = \frac{\partial^2 g_2}{\partial t^2}(z, n)$ are the first and second derivatives evaluated at time $n$, contributing to the underlying dynamics through linear and quadratic functions of time, respectively.

The zeroth, first, or second derivatives provide different "views" of the continuous video dynamics. It is apparent that they share a common latent factor $z$ but not $s$, according to eq. (1). Guiding the model to encode these views into consistent representations in the latent space provides a self-supervised learning strategy for reverse inference of the dynamic latent $z$ rather than the static latent $s$. Therefore, this strategy has the potential to better steer the model to uncover the hidden dynamics from discrete frames through temporal derivatives, which is often missed in traditional frame-based video learning and analysis.

**A thought experiment on physical motion.** Our idea is analogous to and inspired by physical motion. As illustrated in fig. 2, consider a 1-D toy example through a thought experiment . Imagine that we observe the free fall motion of a ball on different planets and aim to infer the planet based on a sequence of snapshots of the ball. The zeroth, first, and second derivatives are the position $y(t)$, velocity $v(t)$, and acceleration $a(t)$, respectively. Evaluating these derivatives at discrete times $n$ gives rise to three sequences, providing different discrete views that collectively expand the continuous motion. Given the physical law, the free fall motion is governed by $y(t) = y_0 + v_0 t - \frac{1}{2}gt^2$, including three latent factors: the initial position $y_0$, the initial velocity $v_0$, and the gravity $g$. It is straightforward to recognize that only the gravity $g$ is involved in the position, velocity, and acceleration. Thus, inferring the representation shared across these different views reveals gravity $g$, which is the defining feature of the dynamics.

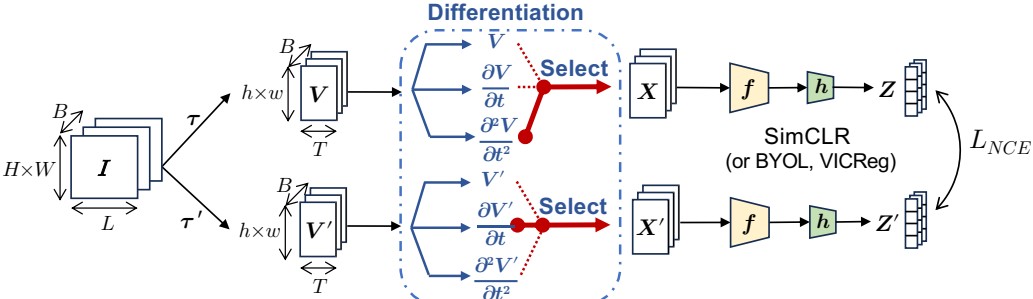

Figure 3: **Illustration of the ViDiDi framework.** For a batch of videos $I$, we do two spatio-temporal augmentations $\tau$ and $\tau'$ to obtain two batches of clips: $V$ and $V'$. These clips are evaluated for the $0^{th}$, $1^{st}$, or $2^{nd}$ order temporal derivatives. Such derivatives are further selected (denoted as $X$ and $X'$) via a balanced alternating learning strategy described in alg. 1. $X$ and $X'$ are the inputs to the video encoder in a 2-stream SSL framework such as SimCLR, BYOL, and VICReg for learning through instance discrimination. $f$ is the video encoder, $h$ is the MLP projection head, and $Z$ and $Z'$ are the encoded embeddings.

## 2.2. The ViDiDi Framework

However, it is non-trivial to extend this intuition to practical learning of dynamics in natural videos. Next, we describe ViDiDi as a learning framework generalizable to multiple existing 2-stream self-supervised video representation methods. It involves 1) creating multiple views from videos through spatio-temporal augmentation and differentiating, 2) a balanced alternating learning strategy for pair-wise encoding of the different views into consistent representations, and 3) plugging this strategy into existing instance discrimination methods, including SimCLR [11], BYOL [12], and VICReg [14]. See fig. 3 for an overview.

**Creating multiple views from videos.** This process contains two steps.

- *Augmentation*: Given a batch of videos $I$ in the shape of $\mathbb{R}^{B \times C \times L \times H \times W}$ where $B$ is the batch size, $L$ is the number of frames per video, $C$ is the number of channels, and $(H, W)$ is the frame size: we sample each video by randomly cropping two clips, each of length $T$. This is a temporal augmentation approach similar to previous works. For each clip, we apply a random set of spatial augmentations [9], including random crops, horizontal flips, Gaussian blur, color jittering, and normalization. In this way, we create two augmented views for every video in the batch, denoted as $V$ and $V'$, each containing a batch of video clips in the shape of $\mathbb{R}^{B \times C \times T \times h \times w}$ where $(h, w)$ is the cropped frame size. See more details in the appendix.

- *Differentiation*: As introduced in section 2.1, we further recover hidden views of clips via temporal differentiation. For every augmented clip within either $V$ or $V'$, we evaluate its $0^{th}$, $1^{st}$, or $2^{nd}$ temporal derivatives. The clips in the same batch will be derived for the same number of times to better support learning of consistent representations across different order of derivatives. In this study, we limit the derivatives up to the $2^{nd}$ order which already shows significant improvements. Denote $V(n) \in \mathbb{R}^{B \times C \times h \times w}$ as a batch of frames selected at time $n$ from the clip $V$, we approximate temporal derivatives with finite forward differences, then the temporal derivatives with respect to $t$ and evaluated at $n$ are:

$$\frac{\partial V}{\partial t}(n) = V(n+1) - V(n) \tag{3}$$

$$\frac{\partial^2 V}{\partial t^2}(n) = V(n+2) - 2 * V(n+1) + V(n) \tag{4}$$

**Balanced alternating learning strategy.** We design a paring schedule for leaning consistent representations among derivatives and original frames for the same video using 2-stream SSL methods.

Specifically, we define seven pairs: $(\boldsymbol{V}, \boldsymbol{V}')$, $(\boldsymbol{V}, \frac{\partial \boldsymbol{V}'}{\partial t})$, $(\frac{\partial \boldsymbol{V}}{\partial t}, \boldsymbol{V}')$, $(\frac{\partial \boldsymbol{V}}{\partial t}, \frac{\partial \boldsymbol{V}'}{\partial t})$, $(\frac{\partial \boldsymbol{V}}{\partial t}, \frac{\partial^2 \boldsymbol{V}'}{\partial t^2})$, $(\frac{\partial^2 \boldsymbol{V}}{\partial t^2}, \frac{\partial \boldsymbol{V}'}{\partial t})$, $(\frac{\partial^2 \boldsymbol{V}}{\partial t^2}, \frac{\partial^2 \boldsymbol{V}'}{\partial t^2})$, each including the temporal derivatives of $\boldsymbol{V}$ and $\boldsymbol{V}'$ evaluated for 0, 1, or 2 times. Such paired derivatives, denoted as $\boldsymbol{X}$ and $\boldsymbol{X}'$, provide inputs to two video encoders in separate streams. At each step (aka batch), we select one pair of derivatives following alg. 1. For each batch, the number of times $\boldsymbol{V}$ and $\boldsymbol{V}'$ each are derived in time, depends on the epoch number as well as additional randomness. Intuitively we choose this strategy to let learning of derivatives guide original frames in a balanced way. So we generally start by pairing the derivatives at higher orders, then the derivatives across different orders, and lastly the derivatives at the zeroth order, and continuing this cycle. We empirically verify that this balanced alternating learning strategy plays an important role in the learning process in the ablation study in appendix B.

---

**Algorithm 1:** Differentiation at Each **Batch**

---

**Data:** epoch $\geq 0$, $(\boldsymbol{V}, \boldsymbol{V}')$
**Result:** $(\boldsymbol{X}, \boldsymbol{X}')$
// Deterministic differentiation step

1 **if** $epoch\%4 = 0$ **then** $(\boldsymbol{X}, \boldsymbol{X}') \leftarrow (\frac{\partial \boldsymbol{V}}{\partial t}, \frac{\partial \boldsymbol{V}'}{\partial t})$ ;

2 **else if** $epoch\%4 = 1$ **then** $(\boldsymbol{X}, \boldsymbol{X}') \leftarrow (\frac{\partial \boldsymbol{V}}{\partial t}, \boldsymbol{V}')$ ;

3 **else if** $epoch\%4 = 2$ **then** $(\boldsymbol{X}, \boldsymbol{X}') \leftarrow (\boldsymbol{V}, \frac{\partial \boldsymbol{V}'}{\partial t})$ ;

4 **else** $(\boldsymbol{X}, \boldsymbol{X}') \leftarrow (\boldsymbol{V}, \boldsymbol{V}')$ ;
// Additional random differentiation step

5 $\epsilon \leftarrow \mathtt{rand}(0, 1)$ ;

6 **if** $\epsilon < 0.5$ **then** $(\boldsymbol{X}, \boldsymbol{X}') \leftarrow (\frac{\partial \boldsymbol{X}}{\partial t}, \frac{\partial \boldsymbol{X}'}{\partial t})$ ;

---

**Plug into existing instance discrimination methods.** Our approach is generalizable to different types of instance discrimination methods, including SimCLR [11] based on contrastive learning, BYOL [12] using teacher-student distillation, and VICReg [14] via variance-invariance-covariance regularization. The above methods are selected as representative examples. By integrating ViDiDi with these different learning objectives, we have trained and tested various models, namely, ViDiDi-SimCLR, ViDiDi-BYOL, and ViDiDi-VIC. We describe ViDiDi-SimCLR below and refer to the appendix for details about other models. $(\boldsymbol{X}, \boldsymbol{X}')$ are two batches of input to the two streams of video encoders, denoted as $f$, which uses a 3D ResNet or other architectures in our experiments. Following the video encoder, $h$ is a multi-layer perceptron (MLP) based projection head, yielding paired embeddings $(\boldsymbol{Z}, \boldsymbol{Z}')$, $Z = [z_1, \ldots, z_B]^T \in \mathbb{R}^{B \times D}$. We evaluate the representational similarity between the $i^{th}$ clip and the $j^{th}$ clip as $s_{i,j} = \boldsymbol{z}_i^\top \boldsymbol{z}_j' / (\|\boldsymbol{z}_i\| \|\boldsymbol{z}_j'\|)$. Given a batch size of $B$, the InfoNCE loss [11] is:

$$\mathcal{L}_{NCE} = \frac{1}{2B} \sum_{i=1}^{B} \log \frac{\exp(s_{i,i}/\alpha)}{\sum_{j=1}^{B} \exp(s_{i,j}/\alpha)} + \frac{1}{2B} \sum_{i=1}^{B} \log \frac{\exp(s_{i,i}/\alpha)}{\sum_{j=1}^{B} \exp(s_{j,i}/\alpha)} \tag{5}$$

## 3. Experiments

### 3.1. Experiment Setup

**Datasets.** We train and evaluate ViDiDi using human action video datasets. **UCF101** [1] includes 13k videos from 101 classes. **HMDB51** [2] contains 7k videos from 51 classes. In addition, we also use larger and more diverse datasets, **K400** [7], aka Kinetics400, including 240k videos from 400 classes, and **K200-40k**, including 40k videos from 200 classes, as a subset of Kinetics 400, helps verify data-efficiency. In our experiments, we pretrain models with UCF101, K400, or K200-40k and then test them with UCF101 or HMDB51, using split 1 for both datasets. **AVA** contains 280K videos from 60 action classes, each video is annotated with spatiotemporal localization of human actions.

**Basics of networks and training.** We use R3D-18 [24] as the default architecture for the video encoder. In addition, we also explore other architectures, namely R(2+1)D-18 [24], MC3-18 [24], and S3D [25]. We test ViDiDi on instance discrimination methods including VICReg, SimCLR, and BYOL. During self-supervised pretraining, we remove the classification head and train the model up to the final global average pooling layer, followed by a MLP-based projection head. We pretrain the model for 400 epochs on UCF101 and K200-40k, and K400. The learning rate follows a cosine decay schedule [26] for all frameworks. A 10-epoch warmup is only employed for BYOL. Weight decay is $1e-6$. We apply cosine-annealing of the momentum for BYOL as proposed in [12]. The temperature for SimCLR is $\alpha = 0.1$, and hyper-parameters for VICReg are $\lambda = 1.0, \mu = 1.0, \nu = 0.05$. We train all models with the LARS optimizer [27] utilizing a batch size of 64 on UCF101 and 256 on Kinetics, with learning rate $\eta = 1.2$. After pretraining, the projection head is discarded.

**Downstream tasks.** We follow the evaluation protocol in previous works [3, 8, 8, 9], including three types of downstream tasks. i) *Video retrieval.* We encode videos with the pretrained encoder, sample videos in the testing set to query the top-k ($k = 1, 5, 10$) closest videos in the training set. The retrieval is successful if at least one out of the $k$ retrieved training videos is from the same class as the query video. ii) *Action recognition.* We add a linear classification head to the pretrained model, and fine-tune this model end-to-end for 100 epoch for action classification. We report the top-1 accuracy. iii) *Action detection.* We follow [9] to finetune the pretrained model as a detector within a Faster R-CNN pipeline on AVA for 20 epochs, and report mean Average Precision (mAP). More details are in appendix D for both pertaining and testing.

## 3.2. Comparisons with Previous Works

Results on both video retrieval (table 2) and action recognition (table 1) suggest that ViDiDi outperforms prior models. Compared to other models trained on Kinetics, ViDiDi-VIC achieves the highest accuracy using K400, while also reaching compatible performance using UCF101 or K200-40k subset for pretraining. Besides, ViDiDi-BYOL achieves the best performance in action recognition on HMDB51, by a significant margin of 5.1% over the recent TCLR method [10]. Importantly, ViDiDi supports efficient use of data. Its performance gain is most significant in the scenario of training with smaller datasets. We discuss this with more details in the following section.

Table 1: **ViDiDi surpasses previous works on action recognition after finetuning.**

| Method | Net | Input | Pretrained | UCF | HMDB |
|---|---|---|---|---|---|
| VCOP [28] | R3D-18 | $16 \times 112$ | UCF101 | 64.9 | 29.5 |
| VCP [29] | R3D-18 | $16 \times 112$ | UCF101 | 66.0 | 31.5 |
| 3D-RotNet [30] | R3D-18 | $16 \times 112$ | K600 | 66.0 | 37.1 |
| DPC [31] | R3D-18 | $25 \times 128$ | K400 | 68.2 | 34.5 |
| VideoMoCo [16] | R3D-18 | $32 \times 112$ | K400 | 74.1 | 43.6 |
| RTT [32] | R3D-18 | $16 \times 112$ | K600 | 79.3 | 49.8 |
| VIE [33] | R3D-18 | $16 \times 112$ | K400 | 72.3 | 44.8 |
| RSPNet [34] | R3D-18 | $16 \times 112$ | K400 | 74.3 | 41.8 |
| VTHCL [22] | R3D-18 | $8 \times 224$ | K400 | 80.6 | 48.6 |
| CPNet [35] | R3D-18 | $16 \times 112$ | K400 | 80.8 | 52.8 |
| CPNet [35] | R3D-18 | $16 \times 112$ | UCF101 | 77.2 | 46.3 |
| CACL [3] | T+C3D | $16 \times 112$ | K400 | 77.5 | - |
| CACL [3] | T+R3D | $16 \times 112$ | UCF101 | 77.5 | 43.8 |
| TCLR [10] | R3D-18 | $16 \times 112$ | UCF101 | 82.4 | 52.9 |
| **ViDiDi-BYOL** | R3D-18 | $16 \times 112$ | UCF101 | **83.4** | **58.0** |
| **ViDiDi-VIC** | R3D-18 | $16 \times 112$ | UCF101 | **82.3** | **53.4** |
| **ViDiDi-VIC** | R3D-18 | $16 \times 112$ | K200-40k | **82.7** | **54.2** |
| **ViDiDi-VIC** | R3D-18 | $16 \times 112$ | K400 | **83.2** | **55.8** |
| VCOP [28] | R(2+1)D-18 | $16 \times 112$ | UCF101 | 72.4 | 30.9 |
| VCP [29] | R(2+1)D-18 | $16 \times 112$ | UCF101 | 66.3 | 32.2 |
| PacePred [36] | R(2+1)D-18 | $16 \times 112$ | K400 | 77.1 | 36.6 |
| VideoMoCo [16] | R(2+1)D-18 | $32 \times 112$ | K400 | 78.7 | 49.2 |
| V3S [37] | R(2+1)D-18 | $16 \times 112$ | K400 | 79.2 | 40.4 |
| RSPNet [34] | R(2+1)D-18 | $16 \times 112$ | K400 | 81.1 | 44.6 |
| RTT [32] | R(2+1)D-18 | $16 \times 112$ | UCF101 | 81.6 | 46.4 |
| CPNet [35] | R(2+1)D-18 | $16 \times 112$ | UCF101 | 81.8 | 51.2 |
| CACL [3] | T+R(2+1)D | $16 \times 112$ | UCF101 | 82.5 | 48.8 |
| **ViDiDi-VIC** | R(2+1)D-18 | $16 \times 112$ | UCF101 | **83.0** | **54.9** |

Table 2: **ViDiDi surpasses previous SSL models on video retrieval.** T + C3D means training with an additional transformer.

| Method | Net | Pretrained | UCF101 | | | HMDB51 | | |
|---|---|---|---|---|---|---|---|---|
| | | | 1 | 5 | 10 | 1 | 5 | 10 |
| SpeedNet [38] | S3D-G | K400 | 13.0 | 28.1 | 37.5 | - | - | - |
| RTT [32] | R3D-18 | K600 | 26.1 | 48.5 | 59.1 | - | - | - |
| RSPNet [34] | R3D-18 | K400 | 41.1 | 59.4 | 68.4 | - | - | - |
| CoCLR [8] | S3D | K400 | 46.3 | 62.8 | 69.5 | 20.6 | 43.0 | 54.0 |
| CACL [3] | T+C3D | K400 | 44.2 | 63.1 | 71.9 | - | - | - |
| **ViDiDi-VIC** | R3D-18 | K200-40k | **49.5** | **63.4** | **71.0** | **24.7** | **45.4** | **56.0** |
| **ViDiDi-VIC** | R3D-18 | K400 | **51.2** | **64.6** | **72.6** | **25.0** | **47.2** | **60.9** |
| VCOP [28] | R3D-18 | UCF101 | 14.1 | 30.3 | 40.0 | 7.6 | 22.9 | 34.4 |
| VCP [29] | R3D-18 | UCF101 | 18.6 | 33.6 | 42.5 | 7.6 | 24.4 | 33.6 |
| PacePred [36] | R3D-18 | UCF101 | 23.8 | 38.1 | 46.4 | 9.6 | 26.9 | 41.1 |
| PRP [39] | R3D-18 | UCF101 | 22.8 | 38.5 | 46.7 | 8.2 | 25.8 | 38.5 |
| V3S [37] | R3D-18 | UCF101 | 28.3 | 43.7 | 51.3 | 10.8 | 30.6 | 42.3 |
| CACL [3] | T+R3D | UCF101 | 41.1 | 59.2 | 67.3 | 17.6 | 36.7 | 48.4 |
| **ViDiDi-VIC** | R3D-18 | UCF101 | **47.6** | **60.9** | **68.6** | **19.7** | **40.5** | **55.1** |

Table 3: **Video Retrieval. ViDiDi exhibits efficient learning and generalizes across different methods.**

| Method | Pretrained | UCF101 | | | HMDB51 | | |
|---|---|---|---|---|---|---|---|
| | | 1 | 5 | 10 | 1 | 5 | 10 |
| SimCLR | UCF101 | 29.6 | 41.4 | 49.3 | 17.5 | 34.7 | 45.1 |
| **ViDiDi-SimCLR** | UCF101 | **38.3** | **54.6** | **64.5** | **17.5** | **38.9** | **52.4** |
| BYOL | UCF101 | 32.2 | 43.0 | 50.5 | 13.8 | 31.1 | 44.4 |
| **ViDiDi-BYOL** | UCF101 | **43.7** | **60.4** | **70.1** | **19.3** | **44.1** | **56.6** |
| VICReg | UCF101 | 31.1 | 43.6 | 50.9 | 15.7 | 33.7 | 44.5 |
| **ViDiDi-VIC** | UCF101 | **47.6** | **60.9** | **68.6** | **19.7** | **40.5** | **55.1** |
| VICReg | K400 | 41.9 | 56.5 | 64.8 | 21.7 | 44.1 | 56.1 |
| **ViDiDi-VIC** | K400 | **51.2** | **64.6** | **72.6** | **25.0** | **47.2** | **60.9** |
| **ViDiDi-VIC** | K200-40k | **49.5** | **63.4** | **71.0** | **24.7** | **45.4** | **56.0** |

## 3.3. Data-efficiency and Generalizability

**Efficient learning with limited data.** ViDiDi learns effective video representations with limited data. As presented in table 2 and table 1, pretrained on small dataset UCF101 or K200-40k, ViDiDi surpasses prior video representation learning works pretrained on large-scale K400 or K600 dataset. Further, ViDiDi-VIC pretrained on UCF101 or K200-40k outperforms its baseline method VICReg pretrained on K400, and also reaches compatible performance as ViDiDi-VIC pretrained on K400, as shown in table 3. This aligns with our intuition that holistic learning of derivatives and original frames can steer the encoder to uncover dynamics features as our intuition discussed in section 2.1. Such a strategy efficiently uncovers dynamic features with limited data and does not rely on a more diverse dataset. To gain insights into how ViDiDi works, we further visualize the spatio-temporal attention section 3.4 , and find that ViDiDi attends to dynamic parts in original frames but VICReg attends to background shortcuts.

Table 4: **Action Detection. ViDiDi exhibits better spatiotemporal action localization.**

| Method | VICReg | **ViDiDi-VIC** | SimCLR | **ViDiDi-SimCLR** | BYOL | **ViDiDi-BYOL** |
|---|---|---|---|---|---|---|
| mAP | 0.089 | **0.106** | 0.079 | **0.094** | 0.087 | **0.118** |

**Generalization across methods, backbones, and downstream tasks.** ViDiDi is compatible with multiple existing frameworks based on instance discrimination. We combine ViDiDi with SimCLR, BYOL, and VICReg and find that ViDiDi improves the performance of video retrieval for each of these frameworks. As shown in table 3, the performance gain is remarkable (up to 16.5%) and consistent across the three frameworks on video retrieval. With fine-tuning, ViDiDi significantly improves the performance (up to 18.0%) over their counterparts without ViDiDi on action recogni-

tion, as shown in fig. 1b. Further, when finetuned on action detection, ViDiDi also exhibits consistent performance gain as illustrated in table 4. ViDiDi is also applicable to different encoder architectures, including R3D-18, R(2+1)D-18, MC3-18, and S3D. As shown in table 5, the performance gain in video retrieval is significant for every encoder architecture tested herein, ranging from 13.7% to 17.0% in terms of top-1 accuracy.

Table 5: **Video Retrieval. ViDiDi generalizes to other backbones.**

| Net | Method | UCF101 | | | HMDB51 | | |
|-----|--------|--------|------|------|--------|------|------|
| | | 1 | 5 | 10 | 1 | 5 | 10 |
| R(2+1)D-18 | VICReg | 30.2 | 44.1 | 51.4 | 15.8 | 33.6 | 45.5 |
| | **ViDiDi-VIC** | **47.2** | **62.6** | **69.8** | **20.6** | **44.1** | **57.7** |
| MC3-18 | VICReg | 31.9 | 44.4 | 51.4 | 15.6 | 35.6 | 46.1 |
| | **ViDiDi-VIC** | **44.1** | **59.8** | **68.0** | **20.3** | **40.3** | **53.4** |
| S3D | VICReg | 29.2 | 41.9 | 49.2 | 12.8 | 29.8 | 40.9 |
| | **ViDiDi-VIC** | **42.9** | **59.0** | **67.5** | **18.4** | **38.4** | **51.4** |

## 3.4. Visualization

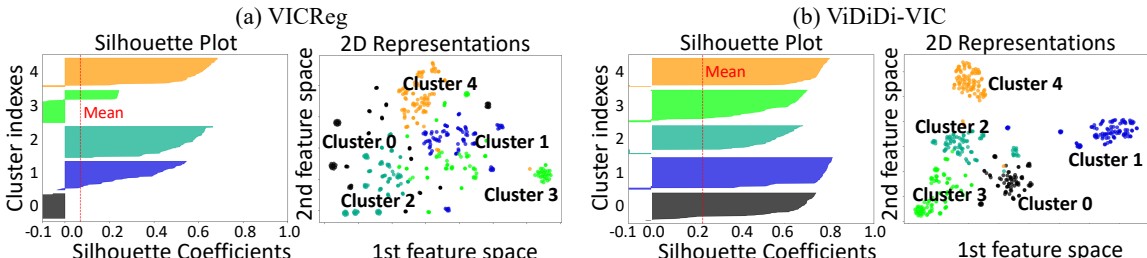

Figure 4: **Silhouette scores and t-SNE of top 5 classes** from VICReg (left) and ViDiDi-VIC (right).

**Clustering in the latent space.** We also examine how video representations from different action classes are distributed in the latent space. We use t-SNE [40] to visualize the representations from five classes. As shown in fig. 4, video representations learned with ViDiDi are better clustered by action classes), showing further separation across distinct classes. Beyond visual inspection, the Silhouette score [41] in table 7 in appendix and fig. 4 quantifies the degree of separation and shows better segregation by video classes with ViDiDi.

**Spatio-temporal attention.** To better understand the model's behavior, we visualize the spatiotemporal attention using Saliency Tubes [42]. ViDiDi leads the model to attend to dynamic aspects of the video, such as motions and interactions, rather than static backgrounds as shown in fig. 5. The model's attention to dynamics is generalizable to videos that the model has not seen. These results align with our intuition that ViDiDi attends to dynamic parts and avoids learning static content as a learning shortcut, resulting in efficient utilization of data as discussed in section 3.3.

## 4. Conclusion

In this paper, we introduce ViDiDi, a novel, data-efficient, and generalizable framework for self-supervised video representation. We utilize the Taylor series to unfold a video to multiple views through different orders of temporal derivatives and learn consistent representations among original clips and their derivatives following a balanced alternating learning strategy. ViDiDi learns to extract dynamic features instead of static shortcuts better, enhancing performance on common video representation learning tasks significantly.

Originating from our intuition of understanding the physical world, there are also parallels between this approach and human vision. The human eyes differentiate visual input into complementary

| Frames | Attn of VIC | Attn of ViDiDi-VIC |
|:------:|:-----------:|:------------------:|

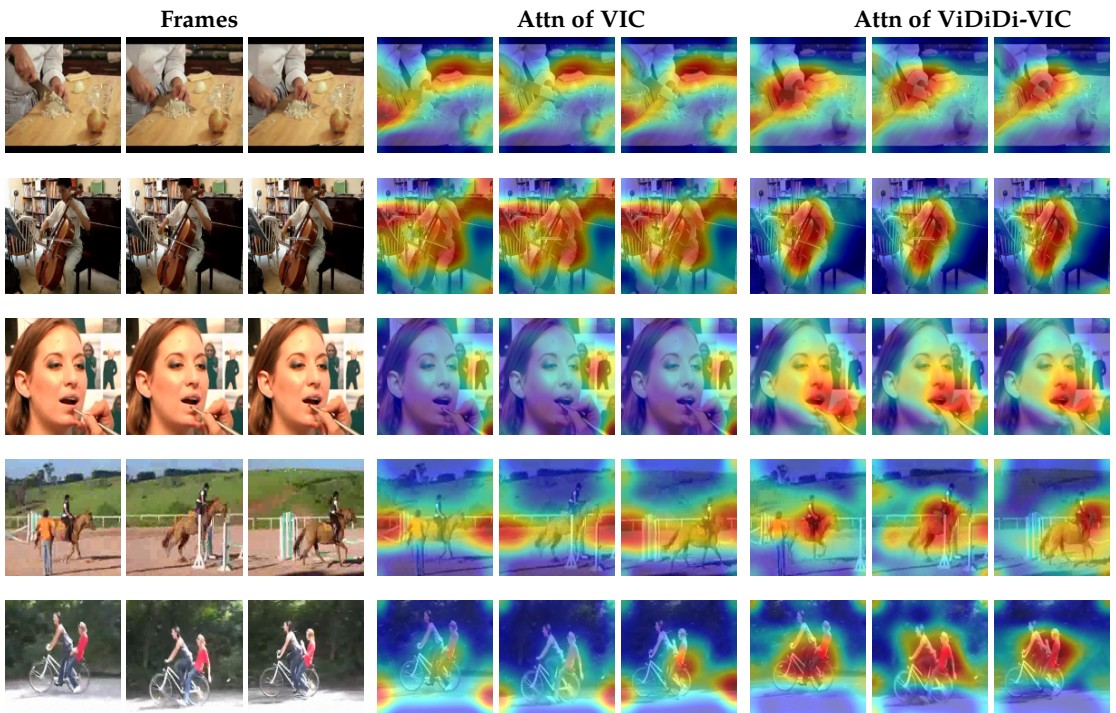

Figure 5: **Spatiotemporal attention on UCF and HMDB51.** Left: Original frames. Middle: Attention from VIC. Right: Attention from ViDiDi-VIC.

retinal views with different spatial and temporal selectivity. The brain integrates these retinal views into holistic internal representations, allowing us to understand the world while being entirely unaware of the differential retinal views. This analogy provides an intriguing perspective on how machines and humans might align their mechanisms in visual processing and learning, which is not explored in the current scope of the paper.

We identify multiple future directions as well. One is to represent different orders of dynamics through different sub-modules in the encoder. In this way, those encoders may teach one another and co-evolve during the learning process, learn different aspects of video dynamics, and be used for different purposes after training. Besides, we can apply Taylor expansion to other modalities or apply ViDiDi to other vision tasks that require more fine-grained understanding of video dynamics [43] such as action segmentation [44]. Also, generative models may create more comprehensive videos or achieve better video editing [45] by learning temporal derivatives as well. Moreover, due to computation limitations, we are unable to scale the model up to state-of-the-art video transformers using high-resolution inputs, which would be worth exploration for large-scale applications. Furthermore, a potentially fruitful direction is to use this approach to learn intuitive physics, supporting agents to understand, predict, and interact with the physical world, since the method's intuition also connects with understanding the physical world.

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

# A. Relationship to Prior Arts

Current methods for SSL of video representations mainly utilize instance discrimination, pretext tasks, multimodal learning, and other ones. In the following, we discuss these results, and highlight the advantages of our method over existing ones.

**Instance discrimination.** It is first applied to images [11–13, 15] and then to videos [3, 8, 9, 16, 17, 22, 23]. Given either images or videos as instances, models learn to discriminate different instances versus different "views" of the same instance, where the views are generated by spatio-temporal augmentations [9, 16]. The learning process is driven by contrastive learning [9], clustering [16], or teacher-student distillation [17]. Recognizing the rich dynamics in videos, some prior works have further modified the loss function to consider each video's temporal attributes, such as play speed [22], time differences [9, 10, 17, 46], frame order [47], and motion diversity [10, 48]. Such modifications do not fully capture the essence of videos as reflections of continuous real-world dynamics, and are usually designed for a specific instance discrimination method. Apart from being applicable to different frameworks of instance discrimination, our approach, is new for its extraction of continuous dynamics by unfolding a video's hidden views via Taylor expansion and temporal differentiation.

**Pretext tasks.** Another category of methods involves creating learning tasks from videos. These tasks have many possible variations, such as identifying transformations applied to videos [30, 32], predicting the speed of videos [34, 36, 38, 39, 49], identifying incorrect ordering of frames or clips [50–53], resorting them in order [28, 54], and solving space-time puzzles [55, 56]. The above methods usually require a complex combination of different tasks to learn general representations, while some recent works utilize large transformer backbones and learn by reconstructing masked areas [57] and further incorporating motion guidance into masking or reconstruction [58, 59]. These tasks provide non-trivial challenges for models to learn but are unlikely to reflect the natural processes through which humans and machines alike may learn and interpret dynamic visual information. In contrast, ViDiDi uses simple learning objectives and models how the physical world can be intuitively processed and understood without the need for complex tasks.

**Others.** In addition, prior methods also learn to align videos with other modalities, such as audio tracks [60–62], video captions [60, 63–65], and optical flows [8, 61, 66–68]. Optical flow also models changes between frames and is related to our method. However, our method is easy to calculate and intuitively generalize to higher orders of motion and guides the learning of the original frames within the same encoder in contrast to an additional encoder for optical flow [8]. Besides, our temporal differentiation strategy may be flexibly adapted to other dynamic data such as audio while optical flow explicitly models the movements of pixels. Incidentally, our proposed *balanced alternating learning strategy* as a simple yet novel way of learning different types of data, may inspire multimodal learning. Some other existing works manipulate frequency content to create augmented views of images [69–71] or videos [66, 72, 73] to make models more robust to out-of-domain data. Related to but unlike these works, we seek a fundamental and computationally efficient strategy to construct views from videos that reflect the continuous nature of real-world dynamics. Besides, apart from a new way of processing data, we propose an alternating learning strategy that is pivotal to boosting learning and has not been explored in previous works.

# B. Ablation Study

ViDiDi involves multiple methodological choices, including 1) the order of derivatives, 2) how to pair different orders of derivatives as the input to two-stream video encoders, and 3) how to prescribe the learning schedule over different pairings. We perform ablation studies to test each design choice.

For the order of derivatives, we consider up to the $2^{nd}$ derivative. For pairing, we consider pairing derivatives in the same order ($1^{st}$ vs. $1^{st}$, $2^{nd}$ vs. $2^{nd}$, etc.) or between different orders ($1^{st}$ vs. $0^{th}$, $1^{st}$ vs. $2^{nd}$, etc.), respectively. For scheduling, we consider either random vs. scheduled selection

of input pairs. With random selection, temporal differentiation is essentially treated as additional data augmentation. In contrast, the scheduled selection (alg. 1) aims to provide a balanced and structured way for the model to learn from various orders of temporal derivatives, where higher order derivatives are intuitively used as guidance of learning the original frames.

As shown in table 6, results demonstrate a progressive improvement in the model's performance in video retrieval, given a higher order of derivatives (from the $1^{st}$ to $2^{nd}$ order), given mixed pairing, and given scheduled selection of input pairs. Therefore, temporal differentiation is not merely another data augmentation trick. Invariance to different orders of temporal derivatives is a valuable principle for SSL of video representations that lead to better performance in downstream tasks. To leverage this principle, it is beneficial to design mixed pairing and prescribe a learning schedule that provides a balanced and holistic view of different orders of temporal dynamics inherent to videos. Details about how we design the groups of models in table 6 are summarized below:

- **Base**: The direct extension of VICReg.
- **+Random** $1^{st}$: Add $1^{st}$ order derivatives as random augmentation.
- **+Random** $1^{st}$ & $2^{nd}$: Add $1^{st}$ and $2^{nd}$ order derivatives as random augmentation.
- **Reverse ViDiDi-VIC**: Reverse the order of pair alternation by epoch in ViDiDi, i.e., line 1-9 in alg. 1.
- **+Schedule** $1^{st}$: Alternate pairs across epochs in the order $(\frac{\partial V}{\partial t}, \frac{\partial V'}{\partial t}) \rightarrow (V, V') \rightarrow (\frac{\partial V}{\partial t}, \frac{\partial V'}{\partial t}) \rightarrow \dots$.
- **+Schedule** $1^{st}$ **& Mix**: switch pairs by epoch in the order $(\frac{\partial V}{\partial t}, \frac{\partial V'}{\partial t}) \rightarrow (\frac{\partial V}{\partial t}, V') \rightarrow (V, V') \rightarrow (\frac{\partial V}{\partial t}, \frac{\partial V'}{\partial t}) \rightarrow \dots$.
- **+Schedule** $1^{st}$ **&** $2^{nd}$ and **+Schedule** $1^{st}$ **&** $2^{nd}$ **& Mix**: Build upon +Schedule $1^{st}$ and +Schedule $1^{st}$ & Mix accordingly with random differentiation at each batch to utilize $2^{nd}$ order derivatives.

Table 6: **Ablation Study.** Video retrieval performance on UCF101 with different design choices.

| Method | UCF101 | | |
|---|---|---|---|
| | 1 | 5 | 10 |
| Base (VICReg) | 31.1 | 43.6 | 50.9 |
| +Random $1^{st}$ | 35.2 | 47.7 | 56.1 |
| +Random $1^{st}$ & $2^{nd}$ | 36.2 | 48.6 | 55.8 |
| Reverse ViDiDi-VIC | 39.1 | 54.7 | 62.9 |
| +Schedule $1^{st}$ | 37.1 | 50.3 | 58.2 |
| +Schedule $1^{st}$ & Mix | 39.3 | 53.2 | 60.9 |
| +Schedule $1^{st}$ & $2^{nd}$ | 40.7 | 56.5 | 64.0 |
| +Schedule $1^{st}$ & $2^{nd}$ & Mix | 43.0 | 59.2 | 66.6 |
| ViDiDi-VIC | **47.6** | **60.9** | **68.6** |

# C. Details of SimCLR, BYOL and VICReg

In this section, we provide more details on how we plug ViDiDi into different instance discrimination frameworks: SimCLR [11], BYOL [12], VICReg [14].

## C.1. Notation

The summary of the SimCLR, BYOL, and VICReg are shown in fig. 6. We begin by introducing the notations. $(X, X')$ represents two batches of input to the discrimination framework, both in the shape $\mathbb{R}^{B \times C \times T \times h \times w}$, containing $B$ clips (or derivatives) of length $T$, and size $h \times w$. $(Z, Z')$ denotes two batches of latents encoded from $(X, X')$, in the shape of $\mathbb{R}^{B \times D}$, containing latents of dimension

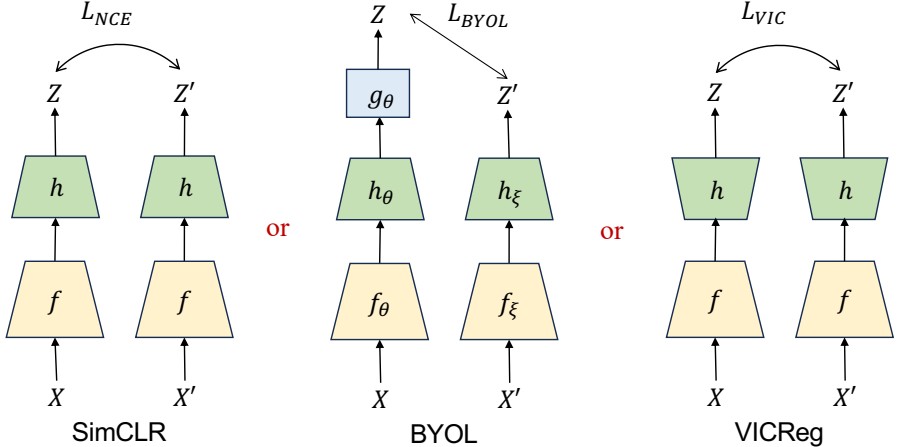

Figure 6: **The SimCLR, BYOL, VICReg details.**

$D$ for $B$ clips. $Z = [z_1, \ldots, z_B]^T$ and $Z' = [z'_1, \ldots, z'_B]^T$, expressed as collects of column vectors. $f$ represents the encoder, which is a 3D convolutional neural network in our experiments. $h$ serves as the projector, either shrinking or expanding output dimensionality. $g$ denotes the predictor. $h$ and $g$ are both realized as multi-layer perceptrons (MLPs). We also introduce the similarity function: $s_{i,j} = z_i^\top z'_j / \left( \|z_i\| \|z'_j\| \right)$.

## C.2. SimCLR

SimCLR [11] is a contrastive learning framework, whose key idea is to contrast dissimilar instances in the latent space. As shown in fig. 6, SimCLR uses a shared encoder $f$ to process $(X, X')$, and then project the output with an MLP projection head $h$ into $(Z, Z')$. $Z = h(f(X))$, $Z' = h(f(X'))$. The InfoNCE loss is defined as:

$$\mathcal{L}_{NCE} = \frac{1}{2B} \sum_{i=1}^{B} \log \frac{\exp(s_{i,i}/\alpha)}{\sum_{j=1}^{B} \exp(s_{i,j}/\alpha)} + \frac{1}{2B} \sum_{i=1}^{B} \log \frac{\exp(s_{i,i}/\alpha)}{\sum_{j=1}^{B} \exp(s_{j,i}/\alpha)} \tag{6}$$

## C.3. BYOL

BYOL [12] is a teacher-student approach. It has an online encoder $f_\theta$, an online projector $h_\theta$, and a predictor $g_\theta$, learned via gradient descent. BYOL uses stop gradient for a target encoder $f_\xi$ and a target projector $h_\xi$, which are updated only by exponential moving average of the online ones $\xi \leftarrow \tau\xi + (1-\tau)\theta$ after each training step, where $\tau \in [0, 1]$ is the target decay rate. $Z = g_\theta(h_\theta(f_\theta(X)))$, $Z' = \mathrm{sg}(h_\xi(f_\xi(X')))$, here sg means stop gradient. The loss is defined as:

$$\mathcal{L}_{BYOL} = \frac{1}{2B} \sum_{i=1}^{B} (2 - 2s_{i,j}) \tag{7}$$

## C.4. VICReg

VICReg [14] learns to discriminate different instances using direct variance, invariance, and covariance regularization in the latent space. It also has a shared encoder $f$ and a shared projector $h$. $Z = h(f(X))$, $Z' = h(f(X'))$. The invariance term is defined as:

$$s(Z, Z') = \frac{1}{B} \sum_{i=1}^{B} \|z_i - z'_i\|_2^2 \tag{8}$$

The variance term constraints variance along each dimension to be at least $\gamma$, $\gamma$ is a constant:

$$v(Z) = \frac{1}{D} \sum_{j=1}^{D} \max\left(0, \gamma - S\left(z^j, \epsilon\right)\right) \tag{9}$$

where S is the regularized standard deviation $S(x, \epsilon) = \sqrt{\mathrm{Var}(x) + \epsilon}$, $\epsilon$ is a small constant, $z^j$ is the $j^{th}$ row vector of $Z^T$, containing the value at $j^{th}$ dimension for all latents in $Z$.

The covariance term constraints covariance of different dimensions to be 0:

$$c(Z) = \frac{1}{D} \sum_{i \neq j} [C(Z)]_{i,j}^2 \tag{10}$$

$C(Z) = \frac{1}{B-1} \sum_{i=1}^{B} \left(z_i - \bar{z}\right)\left(z_i - \bar{z}\right)^T$, $\bar{z} = \frac{1}{B} \sum_{i=1}^{B} z_i$.

The total loss is a weighted sum of invariance, variance, and covariance terms:

$$\begin{aligned}\mathcal{L}_{VIC} =& \lambda s\left(Z, Z'\right) + \mu\left[v(Z) + v\left(Z'\right)\right] \\ &+ \nu\left[c(Z) + c\left(Z'\right)\right]\end{aligned} \tag{11}$$

# D. Implementation Details

## D.1. Augmentation Details

We apply clipwise spatial augmentations as introduced in [9]. All the augmentations are applied before differentiation. For example, for a clip sampled from one video, we do a random crop on the first frame and crop all the other frames in the clip to the same area as the first frame. If a second clip is sampled, we do random crop on its first frame and crop the other frames to the same area. The original frames are extracted and resized to have a shorter edge of 150 pixels. The list of augmentations is as follows:

- Random Horizontal Flip, with probability $0.5$;
- Random Sized Crop, with area scale uniformly sampled in the range $(0.08, 1)$, aspect ratio in $(\frac{3}{4}, \frac{4}{3})$, BILINEAR Interpolation, and output size $112 \times 112$;
- Gaussian Blur, with probability $0.5$, kernal size $(3, 3)$, sigma range $(0.1, 2.0)$;
- Color Jitter, with probability $0.8$, brightness $0.2$, contrast $0.2$, saturation $0.2$, hue $0.05$;
- Random Gray, with probability $0.5$;
- Normalize, mean=$[0.485, 0.456, 0.406]$, std=$[0.229, 0.224, 0.225]$.

## D.2. Network Architecture

The output feature dimension for R3D-18, R(2+1)D-18, and MC3-18 is 512, while 1024 for S3D. In terms of the projector architecture, we use a 2-layer MLP in BYOL, and a 3-layer MLP in Sim-CLR and VICReg, as proposed by [11, 12, 14]. The output dimension of the projector is $d_{BYOL} = 256, d_{SimCLR} = 128, d_{VICReg} = 2048$, and the hidden dimension is $d_{BYOL} = 4096, d_{SimCLR} = 2048, d_{VICReg} = 2048$. The predictor for BYOL is a 2-layer MLP, with output dimension $d = 256$, and hidden dimension $d = 4096$. Batch normalization [74] and Rectified Linear Unit (ReLU) are applied for all hidden layers of projectors and predictors.

## D.3. Pretraining

UCF101, K400, or K200-40k is used as the pertaining dataset. We train the model for 400 epochs on UCF101 or K200-40k, and K400. We set $T = 8$, and select 1 frame every 3 frames. The learning rate follows a cosine decay schedule [26] for all frameworks. The learning rate at $k_{th}$ iteration is $\eta \cdot 0.5 \left[\cos\left(\frac{k}{K}\pi\right) + 1\right]$, where $K$ is the maximum number of iterations and $\eta$ is the base learning rate.

A 10-epoch warmup is only employed for BYOL. Weight decay is set as $1e-6$. We apply cosine-annealing of the momentum for BYOL as proposed in [12]: $\tau = 1 - (1 - \tau_{\text{base}}) \cdot (\cos(\frac{k}{K}\pi) + 1)/2$, and set $\tau_{base} = 0.99$. The temperature for SimCLR is $\alpha = 0.1$, and hyper-parameters for VICReg are $\lambda = 1.0, \mu = 1.0, \nu = 0.05$. We train all models with the LARS optimizer [27] utilizing a batch size of 64 for UCF101 or K200-40k, batch size of 256 for K400, and a base learning rate $\eta = 1.2$. The pretraining can be conducted on 8 GPUs, each having at least 12 GB of memory.

### D.4. Video Retrieval

For the pretrained model without any fine-tuning, we test its performance on video retrieval using nearest-neighborhood in the feature space [3, 8]. Specifically, given a video, we uniformly sample 10 clips of length 16, apply random crop and normalization for data augmentation, encode each clip using the pretrained video encoder, and average the resulting representations into a single feature vector for encoding the given video. Through a nearest-neighborhood model that fits the training set, we use each video in the testing set as a query and retrieve the top-k ($k = 1, 5, 10$) closest videos in the training set. The retrieval is successful if at least one out of the $k$ retrieved training videos is from the same class as the query video. We report the top-k retrieval recall on UCF101 and HMDB51. The retrieval can be conducted on 1 GPU, having at least 24 GB of memory.

### D.5. Action Recognition

We also fine-tune the pretrained model to classify human actions. For this purpose, we add a linear classification head to the pretrained model, and fine-tune it end-to-end on UCF101 or HMDB51 for 100 epochs (see more details in the supplementary material). At training, we sample clips of length 16. We use the SGD optimizer [75] with a momentum value of 0.9. The model is tuned for 100 epochs. The batch size is set at 128, with an initial learning rate of 0.2 which is scaled by $\frac{1}{10}$ at the 60th and 80th epochs. We use a weight decay of $1e-4$. Furthermore, a dropout rate of 0.5 is applied. After fine-tuning, we sample 10 clips of length 16 from each testing video, apply random crop and normalization, feed the results as the input to the fine-tuned model, and average their resulting predictions for the final classification of the video. We report the top-1 action recognition accuracy on UCF101 and HMDB51. The finetuning can be conducted on 8 GPUs, each having at least 12 GB of memory. The testing can be conducted on 1 GPU.

### D.6. Action Detection

We mainly follow the CVRL [9] testing pipeline, taking our pre-trained R3D-18 as the backbone and casting a Faster-RCNN [76] on top of it. To fit the time-sequential nature of the input, we extract region-of-interest (RoI) features using a 3D RoIAlign on the output from the final convolutional block. These features are then processed through temporal average pooling and spatial max pooling. The resulting feature is fed into a sigmoid-based classifier for multi-label prediction. We pretrain our R3D-18 with three different methods(VIC/BYOL/SimCLR) and two different inputs (with/without derivative). We use an AdamW[77] optimizer with a 0.01 learning rate, then shrink the learning rate to half after epoch 5. The dropout rate for Faster-RCNN is 0.5. We perform 20 epochs for our six pretrained weights and run an evaluation after each epoch. We report the epoch with the highest mAP. Our clip length is eight frames with an interval of four frames. The finetuning can be conducted on 2 GPUs, each having at least 48 GB of memory. The testing can be conducted on 1 GPU.

## E. Auxilary Results

### E.1. Silhouette Score

Apart from visualization of clustering in the latent space, we also quantify the clustering using Silhouette Score as illustrated in 7.

Table 7: **Silhouette Score** for Base and ViDiDi with $3, 5, \ldots, 101$ classes. ViDiDi improves the Score, showing better clustering in the latent space.

| Method | Silhouette Score | | | | | |
|---|---|---|---|---|---|---|
| | 3 | 5 | 10 | 15 | 20 | 101 |
| SimCLR | 0.136 | 0.081 | 0.048 | 0.034 | 0.022 | -0.026 |
| **ViDiDi-SimCLR** | **0.210** | **0.132** | **0.096** | **0.078** | **0.058** | **0.003** |
| BYOL | 0.038 | -0.086 | -0.094 | 0.091 | -0.080 | -0.186 |
| **ViDiDi-BYOL** | **0.185** | **0.230** | **0.128** | **0.107** | **0.070** | **0.004** |
| VICReg | 0.110 | 0.069 | 0.044 | 0.036 | 0.017 | -0.038 |
| **ViDiDi-VIC** | **0.235** | **0.232** | **0.150** | **0.138** | **0.098** | **0.014** |

(a) VIDReg (left) and ViDiDi-VIC (right).

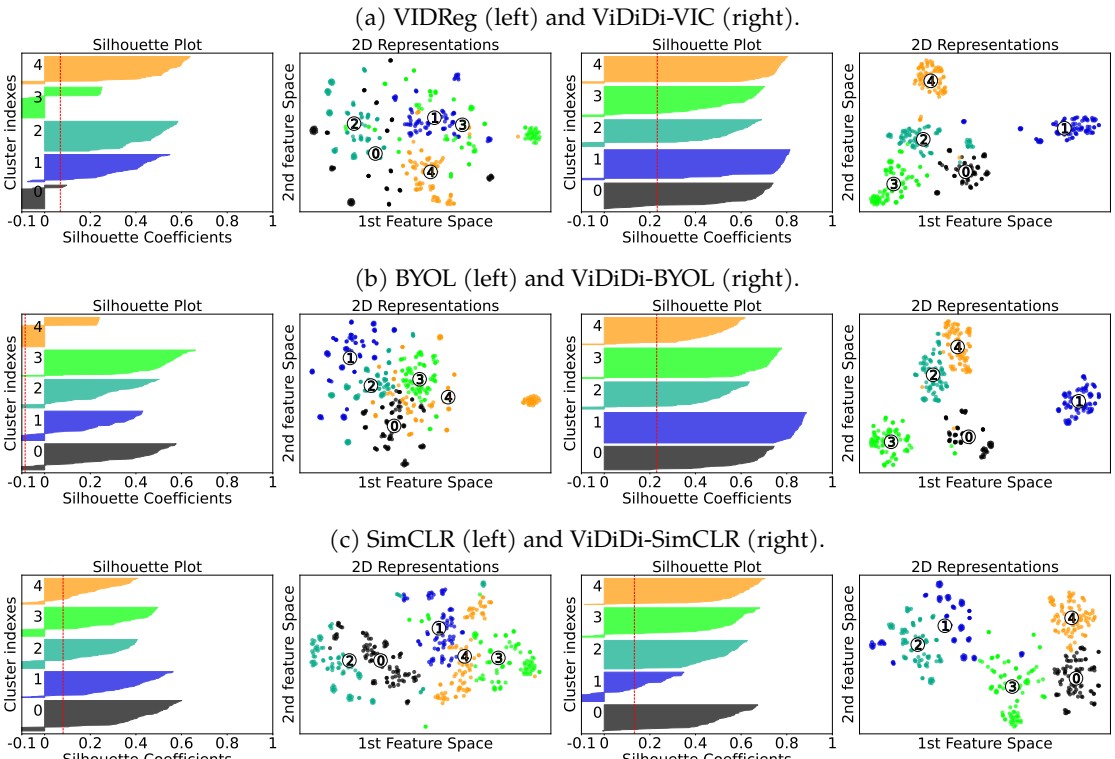

(b) BYOL (left) and ViDiDi-BYOL (right).

(c) SimCLR (left) and ViDiDi-SimCLR (right).

Figure 7: **Silhouette scores and t-SNE plots of top 5 classes in UCF101 train**.

## E.2. Clustering of Latent Space

We provide more visualization of the clustering phenomenon for VICReg, BYOL, and SimCLR, with or without the ViDiDi framework; on UCF101 train dataset or test dataset; utilizing 5 or 10 classes of videos. Here, for each model, we choose the top 5 or 10 classes of videos that are best retrieved during the video retrieval experiments. The results are shown in fig. 7, fig. 8, fig. 9, and fig. 10. ViDiDi provides consistently better clustering in the latent space for both train data and test data.

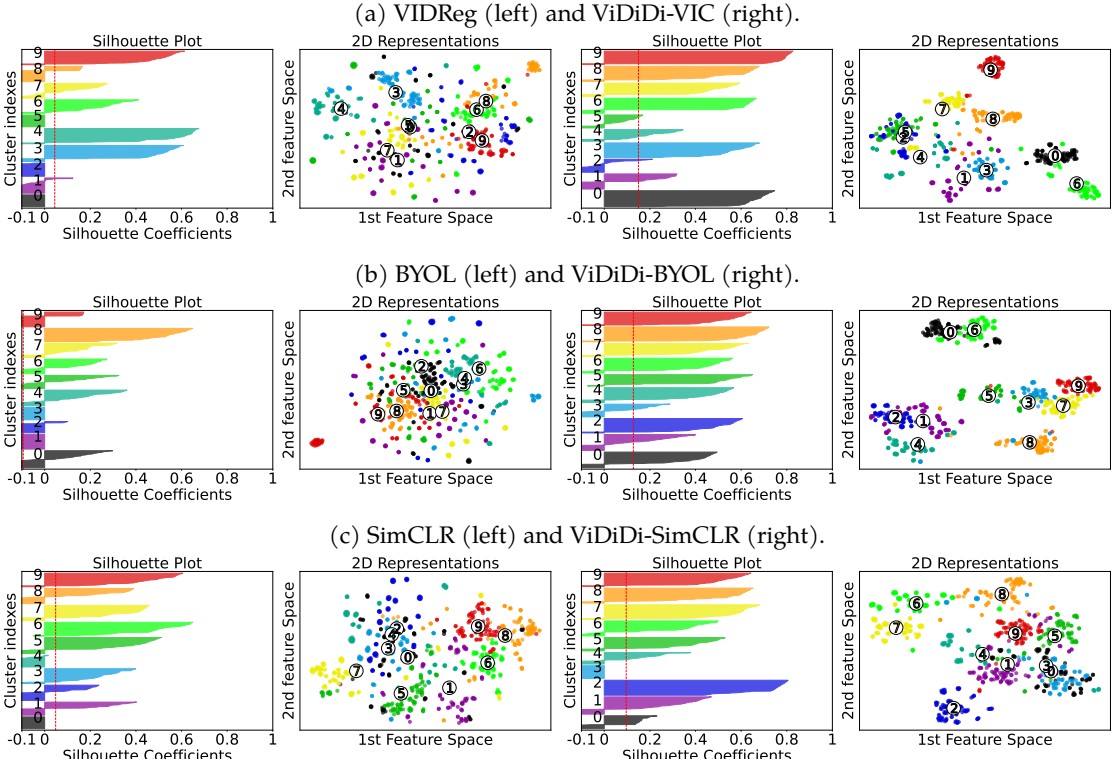

Figure 8: **Silhouette scores and t-SNE plots of top 10 classes in UCF101 train.**

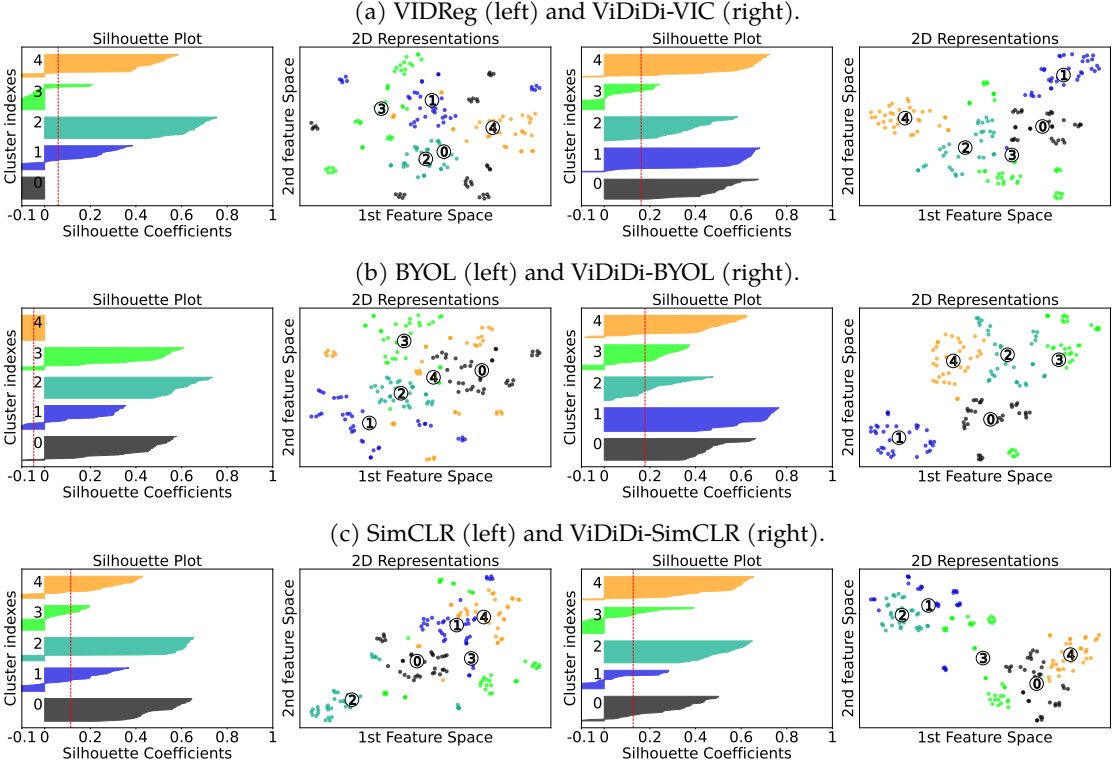

Figure 9: **Silhouette scores and t-SNE plots of top 5 classes in UCF101 test.**

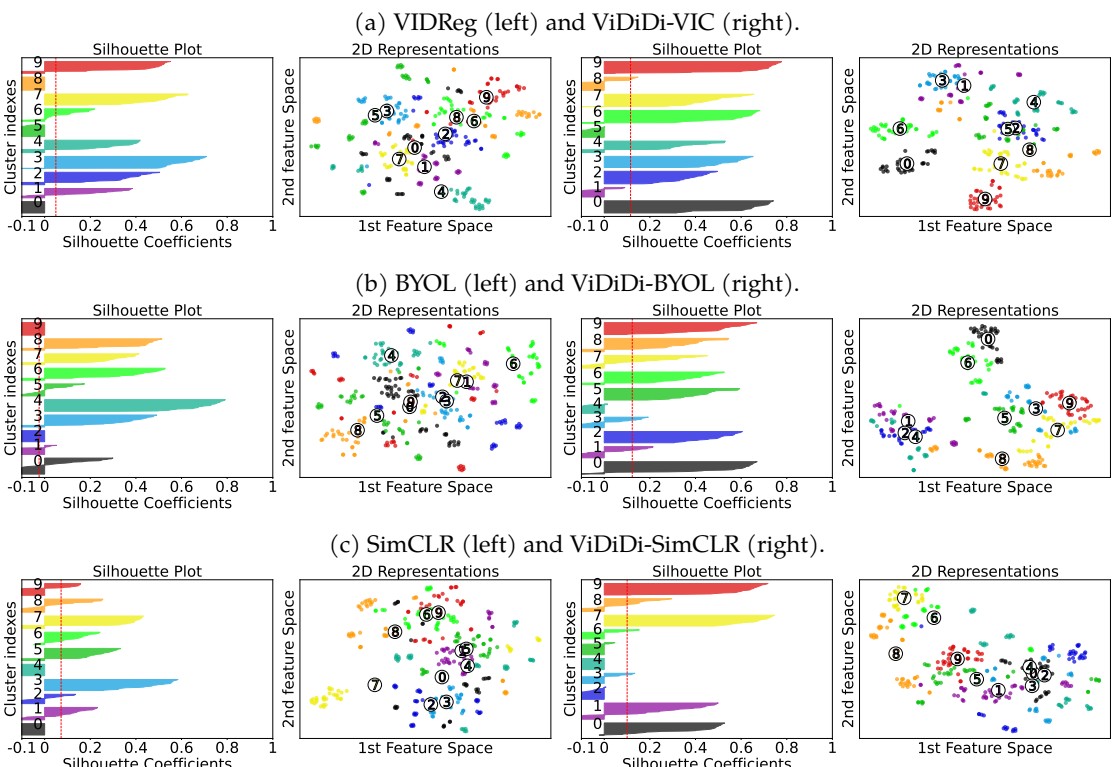

Figure 10: **Silhouette scores and t-SNE plots of top 10 classes in UCF101 test.**

## E.3. Spatio-temporal Attention

We provide more visualization of the attention for VICReg, BYOL, and SimCLR, with or without the ViDiDi framework; on UCF101 dataset or HMDB51 dataset. The results are presented in fig. 11, fig. 12, fig. 13, fig. 14, and fig. 15.

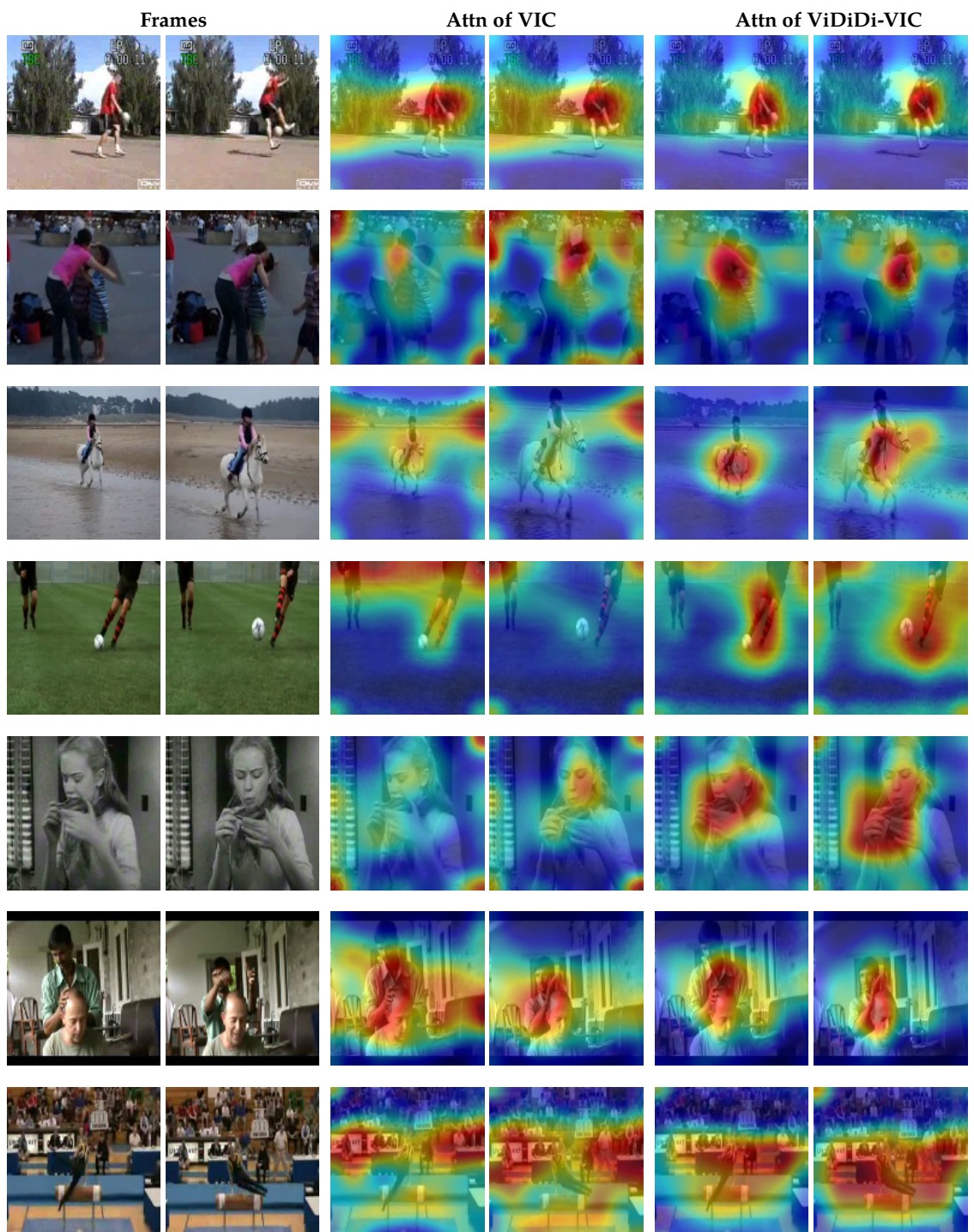

Figure 11: **More spatiotemporal attention for VICReg and ViDiDi-VIC.** Left: Original frames. Middle: Attention from VIC. Right: Attention from ViDiDi-VIC.

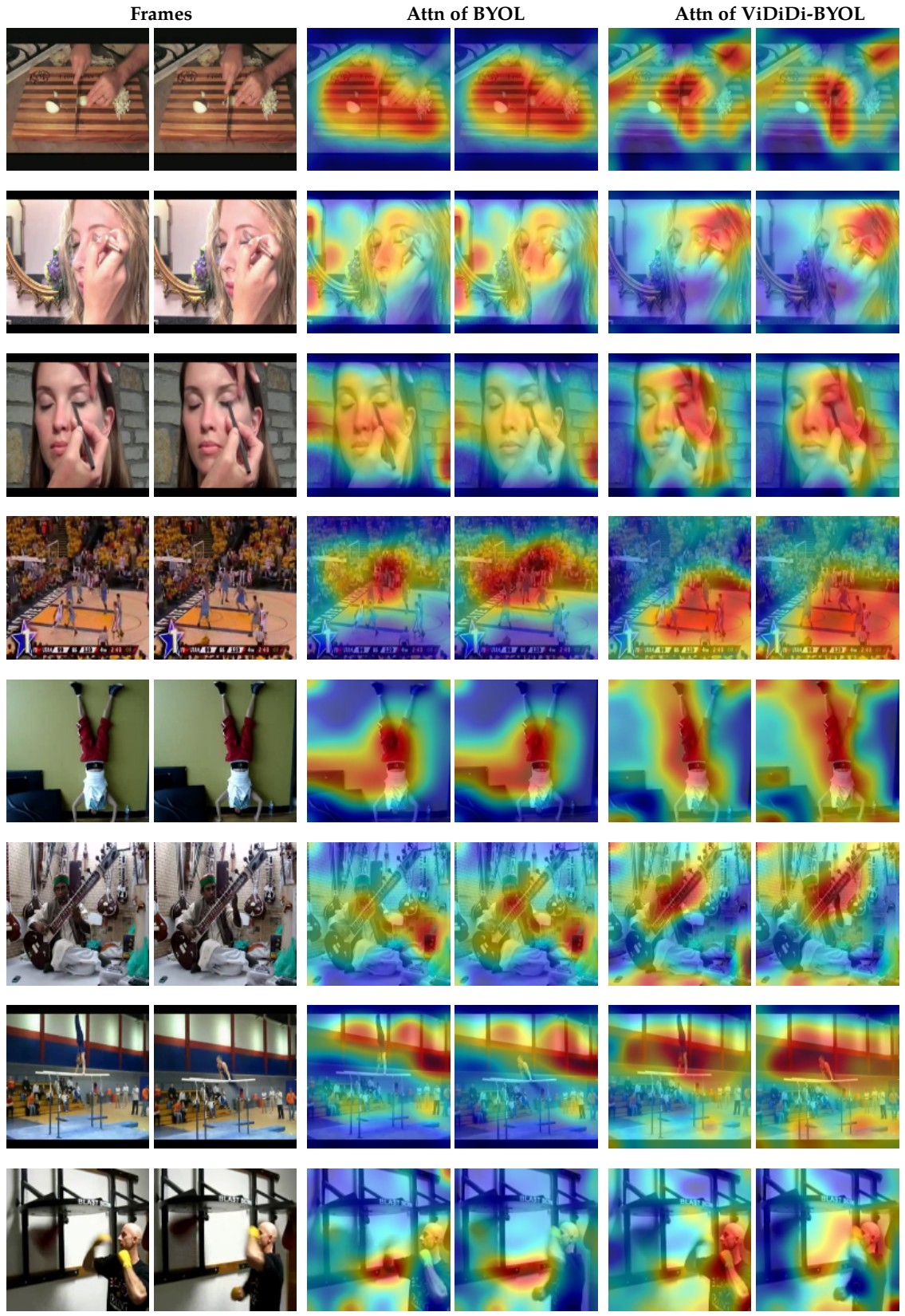

Figure 12: **Spatiotemporal attention on UCF101.** Left: Original frames. Middle: Attention from BYOL. Right: Attention from ViDiDi-BYOL.

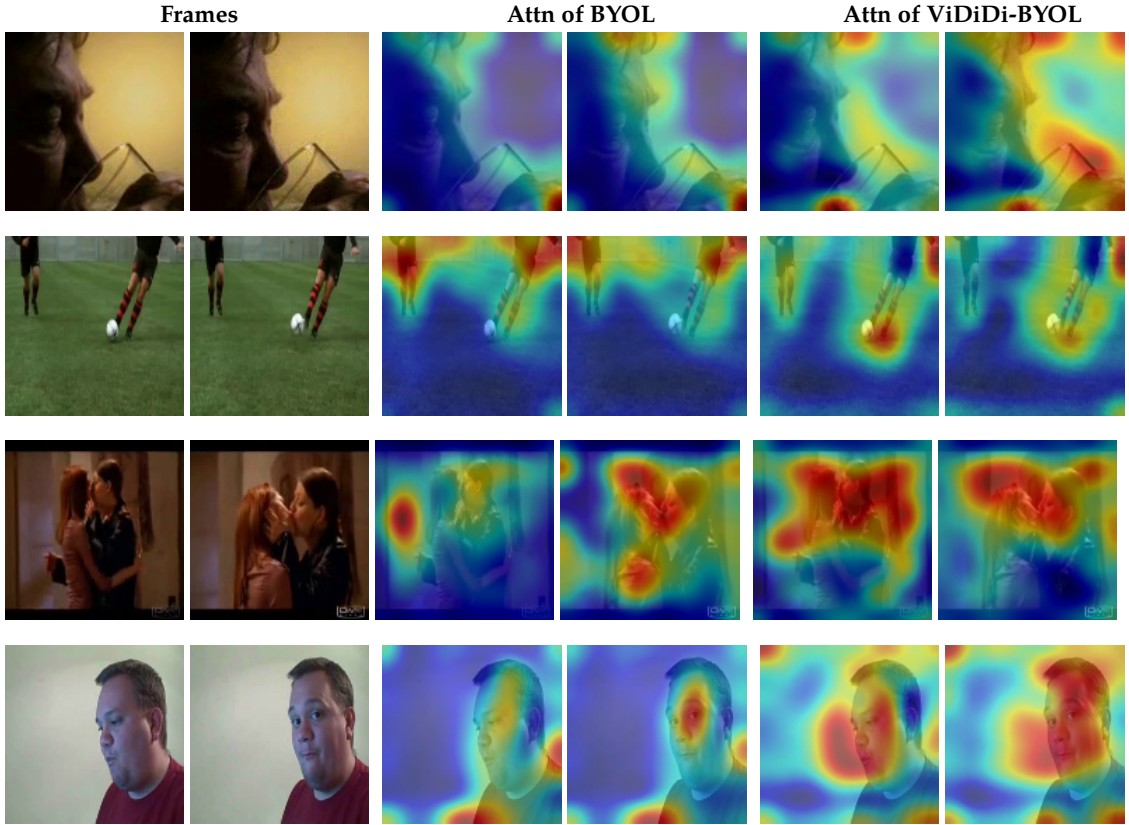

Figure 13: **Spatiotemporal attention on HMDB51.** Left: Original frames. Middle: Attention from BYOL. Right: Attention from ViDiDi-BYOL.

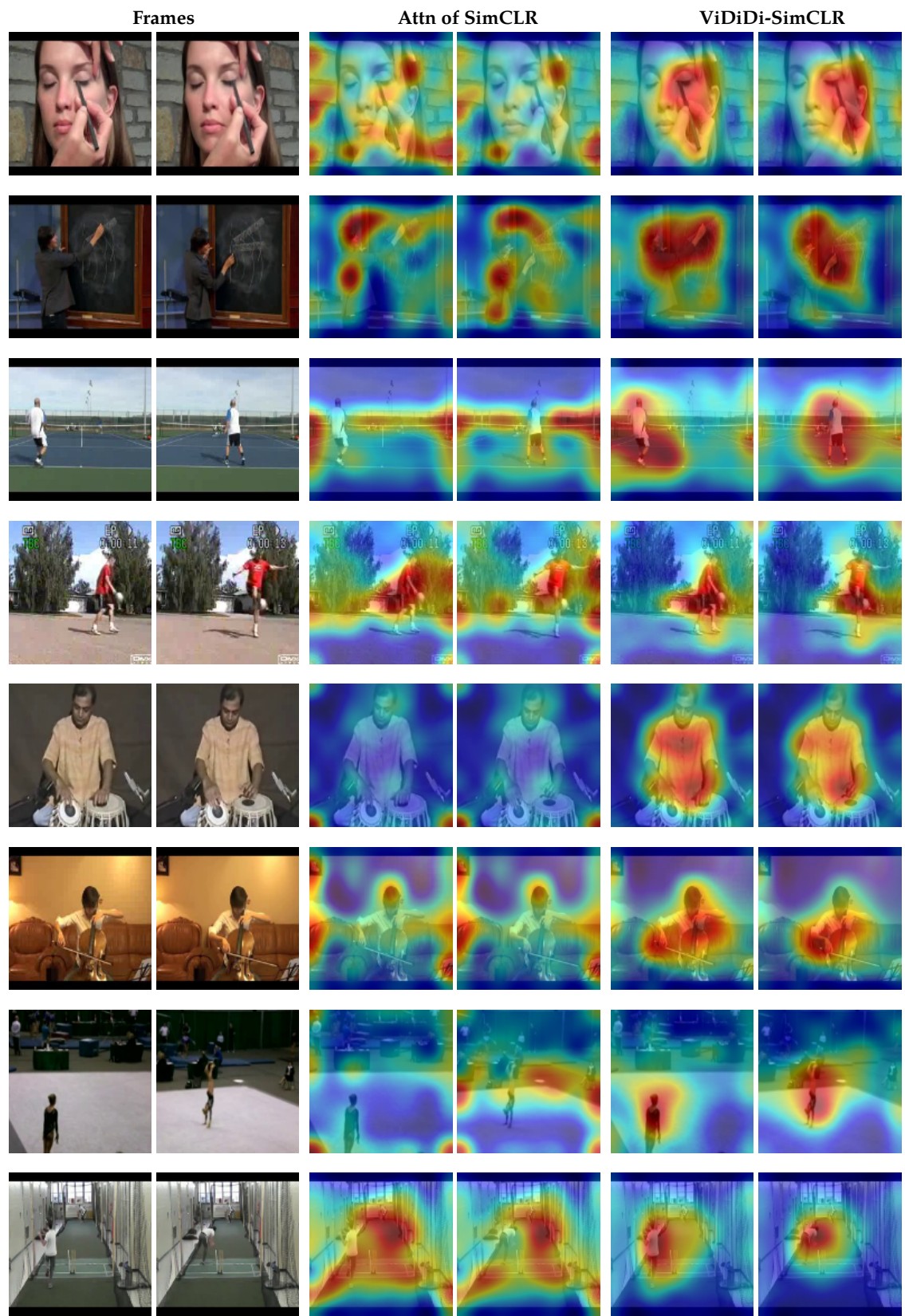

Figure 14: **Spatiotemporal attention on UCF101.** Left: Original frames. Middle: Attention from SimCLR. Right: Attention from ViDiDi-SimCLR.

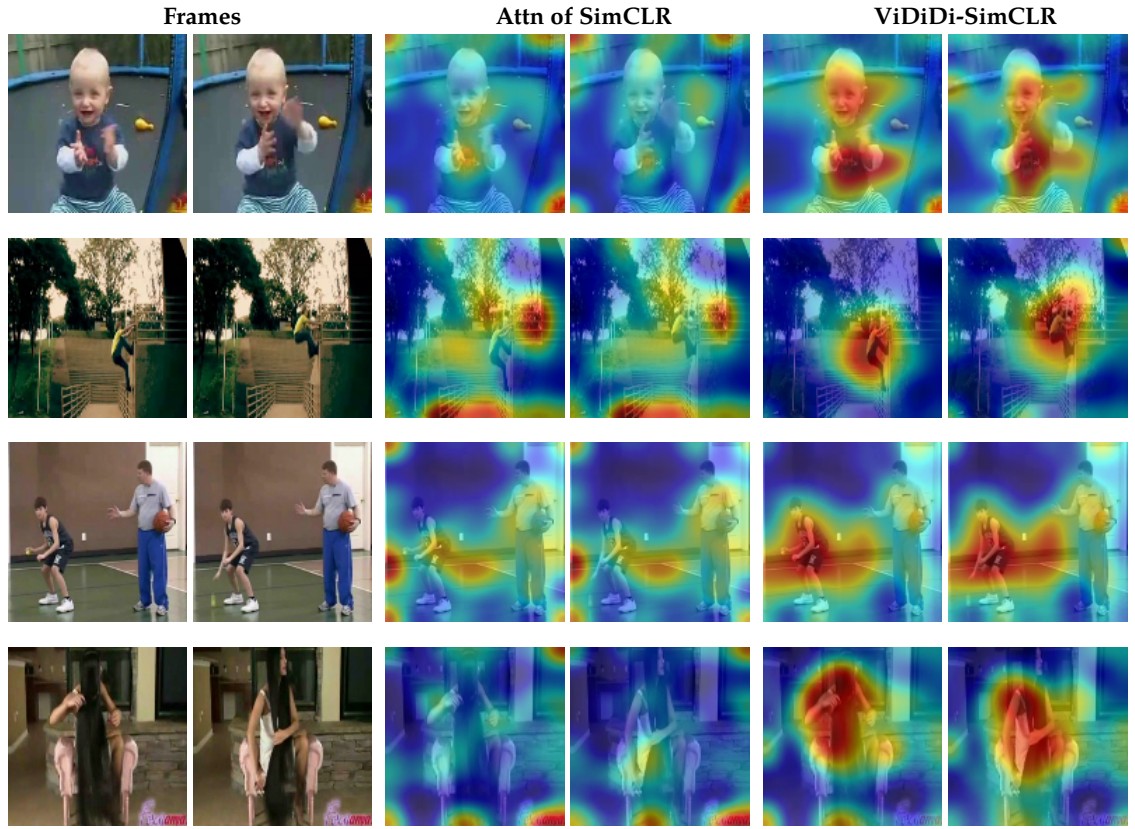

Figure 15: **Spatiotemporal attention on HMDB51.** Left: Original frames. Middle: Attention from SimCLR. Right: Attention from ViDiDi-SimCLR.

