# OpenReview forum: "Enhancing Video Representation Learning with Temporal Differentiation"
_CPAL.cc/2025/Proceedings_Track — CPAL 2025 (Proceedings Track) Poster_

### Official Review · Reviewer_NrYY · 2025-01-11
**Creative approach for video SSL**

**Rating:** 7
**Confidence:** 3

**Review:**

This paper introduces a novel self-supervised method for video representation —ViDiDi (Video Time-Differentiation for Instance Discrimination). This method incorporates temporal derivatives of its frame sequence into instance discrimination frameworks (e.g., SimCLR, BYOL, VICReg).  Using this strategy, the encoder emphasizes dynamic features over static content. The dynamic features might be the shared feature across these different views, like gravity, which is involved in position, velocity, and acceleration. Consequently,  if the instance discrimination framework is applied to learn from different-order temporal derivatives, they might be able to understand the common driven force.  The author ran multiple experiments to demonstrate that ViDiDi provides substantial improvement across benchmarks for video retrieval, action recognition, and action detection while using limited data. The choice of benchmark as well as the baseline model they compare to looks reasonable to me.
It seems that ViDiDi can also be viewed as a type of temporal augmentation. By introducing temporal derivatives, ViDiDi creates multiple views of the same video, similar to traditional augmentations like cropping or flipping. This might be helpful for the model to learn a robust representation. It would be nice if the author could add more comparisons between other augmentation methods.

The author used ball falling as a thought experiment to explain the physical intuition behind this method. This was an interesting and creative way to convey the physical intuition. While the intuition is compelling, I wonder can the authors clarify how would they apply this to real-world video data—particularly at the pixel level. A single video frame may contain multiple objects, which could appear, disappear, or reappear across frames. In addition, it would be helpful for the authors to provide more explanation on how their model interprets latent factors s and z in such complex scenarios. What could be the feature represented by the latent factors? Individual objects, global scene dynamics, or some other features?

I wonder could the author quantify the practical impact of each derivative order (e.g., second-order acceleration) on downstream tasks. This might help the reader to understand better the unique contributions of higher-order derivatives and their specific roles in video representation learning. In the Algorithm 1
Pseudocode, I did not see anywhere that the author mention the second order derivative. It would be nice if the authors could include how does the inclusion of the second-order derivative contribute to tasks like action detection compared to using only zeroth- and first-order derivatives.

Overall, this paper presents an creative approach based on physical intuition to improve current SSL method with strong empirical result.

---

### Official Review · Reviewer_yBtL · 2025-01-13

**Rating:** 5
**Confidence:** 4

**Review:**

Summary:

The manuscript introduces ViDiDi (Video Time-Differentiation for Instance Discrimination), which is a self-supervised learning method for video representation learning that emphasizes motion features over static backgrounds. The method uses temporal derivatives of video frames to capture higher-order motion dynamics, which is integrating these derivatives into a Taylor series expansion. The results show that ViDiDi, when integrated into existing frameworks like VICReg, BYOL, and SimCLR, it enhances performance on benchmarks such as 1) video retrieval, 2) action recognition, and 3) detection without using large models or huge datasets. The proposed work also highlights applications in improving dynamic feature-based video representations.

Strengths:
The paper introduces a new approach to representing continuous video dynamics using different orders of temporal derivatives inspired by the Taylor series expansion. This perspective effectively emphasizes motion features and uncovers hidden dynamics in video frames. The proposed method gives a general self-supervised dynamics learning framework that ensures consistency among temporal derivatives using a balanced alternating learning strategy.

The paper shows the data efficiency and performance improvements of the method on standard video representation learning tasks, including video retrieval, action recognition, and detection. It also provides insights into the learned dynamic features through attention maps and subspace clustering visualizations, which is able to validate the framework's effectiveness.

Weaknesses:

In Section 2.2, the method mentions about the 1) augmentation and 2) differentiation as creating multi-view for videos. However, it is not clear to me and I wonder if the authors could give more intuition about the reasons why these two strategies work for the tasks in the paper. Clarification is needed these two strategies especially for differentiation.

It is also not clear that the authors use only SimCLR, BYOL, and ViCReg for the ablation of self-supervised learning. I did not see any sections which talks about the difference between them (e.g., with / without using negative pairs). Also, why not using other methods (e.g., MoCoV2..etc)

---

### Official Review · Reviewer_tPs8 · 2025-01-15
**In this paper, the authors propose a new representation learning method for video from a new perspective, which is interesting by utilizing the Taylor series. Different from existing video representation methods, the proposed method pays attention to the underlying dynamics among different orders of temporal differentiation.**

**Rating:** 6
**Confidence:** 3

**Review:**

Strengths:
(1)The proposed method is generalizable and data-efficient, which can be appled to self-supervised video representation learning.

(2)A new perspective of using the Taylor series expansion to obtain the temporal derivatives at each frame.

(3)The experimental results are promising across different datasets, showing the strengths of exploring both the zeroth-order derivative, first-order and second-order derivative.

Weaknesses:
(1)In algorithm 1, why 4 is selected to perform different operations?what is the rationale behind it? It seems that the authors overlook this problem.

(2) The choice to limit temporal derivatives to the second order seems arbitrary without proper justification or ablation studies exploring higher orders. Why only zeroth-order derivative, first-order and second-order derivative are selected? The authors should provide relative ablation study or theoretical justication.

(3)The models compared in Tables 1 and 2 use different pretraining datasets, making direct performance comparisons less reliable. Consistent pretraining setups across all models should be enforced.

---

### Meta-Review · Area_Chair_Z251 · 2025-02-04

**Recommendation:** Accept (Poster)
**Confidence:** 3

**Metareview:**

This work introduces ViDiDi (Video Time-Differentiation for Instance Discrimination), a novel self-supervised learning (SSL) framework for video representation learning. Inspired by physical motion, ViDiDi leverages temporal derivatives (e.g., velocity and acceleration) to extract dynamic features from videos, improving performance on tasks like action recognition, video retrieval, and action detection. It integrates seamlessly into existing SSL methods like SimCLR, BYOL, and VICReg and is particularly effective in data-efficient scenarios.

Inspired from Taylor series, videos are treated as continuous dynamic processes, represented through zeroth-order (frames), first-order (velocity), and second-order (acceleration) derivatives. Temporal derivatives emphasize motion dynamics, enabling the model to focus on dynamic features rather than static backgrounds.  ViDiDi is adaptable to multiple SSL frameworks and encoder architectures. And the method achieves significant performance gains even with smaller datasets, improving data efficiency.

Despite the mixed opinions among the reviewers, I slightly recommend accepting this paper. The primary concerns raised by the reviewers pertain to the need for a more comprehensive explanation, evaluation, and introduction of important concepts and algorithms. However, the key contributions of the paper are well-supported by the reviewers.


Reviewer tPs8 question several points on the details fo the algorithm, the choice of orders of the Taylor series, and fairness of the experiments. The author responds accordingly but is still suggested to improve the presentation and adding more necessary details. Reviewer yBtL raises the questions of choosing baselines  in ablation study and intuition of the underlying intuition of technical augmentation and differentiation, and Reviewer NrYY asked about practical impact of each derivative order on downstream tasks and more physical intuition. It is highly recommended the authors add more explanations and intuitions accordingly.

---

### Decision · Program_Chairs · 2025-02-11

Accept (Poster)